DOI: 10.1038/s41467-018-05799-w | OPEN

# Enhanced photon collection enables four dimensional fluorescence nanoscopy of living systems

Luciano A. Masullo [1], Andreas Bodén[1], Francesca Pennacchietti[1], Giovanna Coceano[1], Michael Ratz[2] & Ilaria Testa [1]

The theoretically unlimited spatial resolution of fluorescence nanoscopy often comes at the expense of time, contrast and increased dose of energy for recording. Here, we developed MoNaLISA, for Molecular Nanoscale Live Imaging with Sectioning Ability, a nanoscope capable of imaging structures at a scale of 45–65 nm within the entire cell volume at low light intensities (W-kW cm$^{-2}$). Our approach, based on reversibly switchable fluorescent proteins, features three distinctly modulated illumination patterns crafted and combined to gain fluorescence ON–OFF switching cycles and image contrast. By maximizing the detected photon flux, MoNaLISA enables prolonged (40–50 frames) and large (50 × 50 μm$^2$) recordings at 0.3–1.3 Hz with enhanced optical sectioning ability. We demonstrate the general use of our approach by 4D imaging of organelles and fine structures in epithelial human cells, colonies of mouse embryonic stem cells, brain cells, and organotypic tissues.

[1] Department of Applied Physics and Science for Life Laboratory, KTH Royal Institute of Technology, 100 44 Stockholm, Sweden. [2] Department of Cell and Molecular Biology, Karolinska Institutet, 17176 Stockholm, Sweden. These authors contributed equally: Luciano A. Masullo, Andreas Bodén, Francesca Pennacchietti. Correspondence and requests for materials should be addressed to I.T. (email: ilaria.testa@scilifelab.se)

Observing the interplay of organelles and macromolecular complexes inside living cells and tissues calls for the continuous development of minimally invasive optical systems performing at high spatio-temporal resolution. Nowadays, the spatial resolution of fluorescence nanoscopy approaches the nanoscale (10–50 nm) by optically controlling the ability of molecules to fluoresce either in a deterministic or stochastic fashion[1–5]. However, the current approaches to fluorescence nanoscopy, even if powerful, are often limited by high doses of light, low contrast, small fields of view or slow recording times. The problem of high illumination doses was partially overcome using approaches like Reversible Saturable OpticaL Fluorescent Transition (RESOLFT)[6–8] by using reversibly switchable fluorescent proteins (rsFPs)[9–12]. Here, the coordinate targeted fluorescence ON–OFF switching of the rsFP requires intensities in the range of W-kW cm$^{-2}$ to produce images with sub-100 nm spatial resolution. Modern wide-field (WF) RESOLFT implementations[13,14] can reach relatively fast acquisitions of large fields of view. However, WF-RESOLFT imaging is mostly limited to bright cellular structures in 2D. This limitation stems from the fact that the uniform illumination used to switch to the ON state and to read-out the rsFP causes unnecessary switching and generates signal from out-of-focus planes of the specimen, which hampers the image contrast in 3D samples. Furthermore, even the signal generated by adjacent emitting spots in the focal plane is severely affected by crosstalk, especially in a highly parallelized implementation.

Other approaches such as non-linear structured illumination microscopy[15–17] and its recent implementation featuring Patterned Activation[18] also minimizes the illumination dose if applied to rsFPs[19]. Here, the super resolution information is encoded in the frequency space of the image and therefore has to be extracted through image processing, which is prone to artifacts[20]. This is especially relevant in dim structures with moderately low signal to noise ratio (SNR), such as in cells exhibiting endogenous levels of rsFP fusion expression and in 3D samples where out-of-focus background dominates. A nanoscope able to record robust raw data rapidly and with sub-100 nm spatial resolution across the entire 3D space of cells and tissues is still missing.

To overcome these limitations we developed Molecular Nanoscale Live Imaging with Sectioning Ability (MoNaLISA). This nanoscope features light patterns with optimized shape and periodicities to switch ON, OFF and read out the fluorescence of the rsFPs. To efficiently switch the molecule into the OFF state with a minimal light dose we choose a small periodicity in order to achieve sharp intensity zeros. On the other hand, the ON-switching and read-out patterns are based on multi-spot arrays with larger periodicity in order to maximize the photon collection and minimize switching fatigue and detection cross-talk. Overall, a configuration of light patterns with distinctly different periodicities enable to image structures in the entire cell at 45–65 nm spatial lateral resolution thanks to both optical sectioning and higher photon collection.

## Results

### Basic concept.
The MoNaLISA imaging is performed with the progression of three light illuminations for ON-switching, OFF-switching, and read-out of the rsFPs (see Fig. 1a, b). Each illumination step is modulated in space. Both ON-switching and read-out are composed of $N$ individual foci[21,22], separated by the same multi-foci periodicity $P_{MF}$. The OFF-switching pattern, responsible for the sub-diffractive resolution, features standing waves with multiple intensity minima at a periodicity $P_{SW}$. By aligning these minima with the multi-foci maxima, the emitting spots are confined beyond the diffraction limit giving the super resolution ability.

The independently generated light patterns allow choosing the smallest $P_{SW}$ in order to reduce the light doses of the OFF pattern (see Supplementary Figure 2) and the optimal $P_{MF}$ to maximize the photon collection as well as the sectioning. In fact, the fluorescence emitting spots are well separated only when $P_{MF}$ is larger than the detection point-spread-function (see Fig. 1c). On the contrary, a small $P_{MF}$ leads to a much closer distance between adjacent emitting regions (see Fig. 1d), which causes severe crosstalk (see Fig. 1e). Considering instead the distinctly separated fluorescence foci of MoNaLISA, we can record most of the signal by applying an optimal digital pinhole with a FWHM of ~$\lambda$/2NA. Thus, we maximize the photon collection efficiency without compromising the quality of the image (see Fig. 1f, green dotted line). In WF-RESOLFT since $P_{MF}$ and $P_{SW}$ are coupled, highly parallelized interference OFF patterns will result in closer emission foci. Hence, the extraction of information requires using either a very small digital pinhole in the detection or intensive computational unmixing. These will cause a loss of signal (see Fig. 1d, gray area) eventually resulting in images with low SNR or possible artifacts due to error-prone reconstruction algorithms. As an example, digitally pinholing with a Gaussian FWHM of 50 nm, as used in previous WF-RESOLFT implementations, led to more than 60% loss in SNR (see Fig. 1f, gray dotted line, and Supplementary Note 1). Numerical simulations of the imaging systems (see Fig. 1i and Supplementary Figure 9), as well as comparative experiments (see Supplementary Figure 1) demonstrate the contrast gain of MoNaLISA. The image quality is further enhanced by saving ON–OFF switching fatigue and by the optical sectioning. Indeed, the multi-foci ON-switching of MoNaLISA leads to a spatially confined protein population able to interact with the OFF-switching light.

This way the majority of the proteins are kept in the OFF state and only a few of them are exposed to the intensity maxima of the ON and OFF-patterns in the same cycle, which is extremely important to minimize unwanted cycles and photobleaching. In WF-RESOLFT the entire population of rsFPs are repeatedly exposed to peak intensities of both the ON and OFF-switching light causing unnecessary cycling, especially in 3D.

In addition to saving ON–OFF cycles and enhancing the detection efficiency, the use of multi-foci patterns for ON-switching and read-out enables confocal optical sectioning and sub-100 nm lateral resolution simultaneously. Two factors contribute to the sectioning ability of the nanoscope. First, the confinement of the signal generated by focused light is further enhanced by the progressive rsFPs absorption of the focused ON-switching and excitation light, which leads to a quadratic dependence in the intensities (see Supplementary Note 4). Second, digital pinholing in detection rejects potential out of focus emission further improving the sectioning ability of the system.

### Nanoscale imaging in cells endogenously expressing rsEGFP2.
Our experimental implementation of MoNaLISA featured $N$ = 4356 foci generated with two square microlens arrays (see Fig. 1j). We chose $P_{MF}$ = 750 nm, $P$sw = 250 nm, and a FWHM of 250 nm for the digital Gaussian pinhole. These values were carefully chosen in order to achieve optimal results in terms of optical sectioning (see Supplementary Note 5–6 and Supplementary Figure 3–8). For the rsFPs employed in these experiments we used 405 nm and 488 nm light for the ON-switching and readout multi-foci patterns, respectively. The OFF-switching pattern was also generated with 488 nm light.

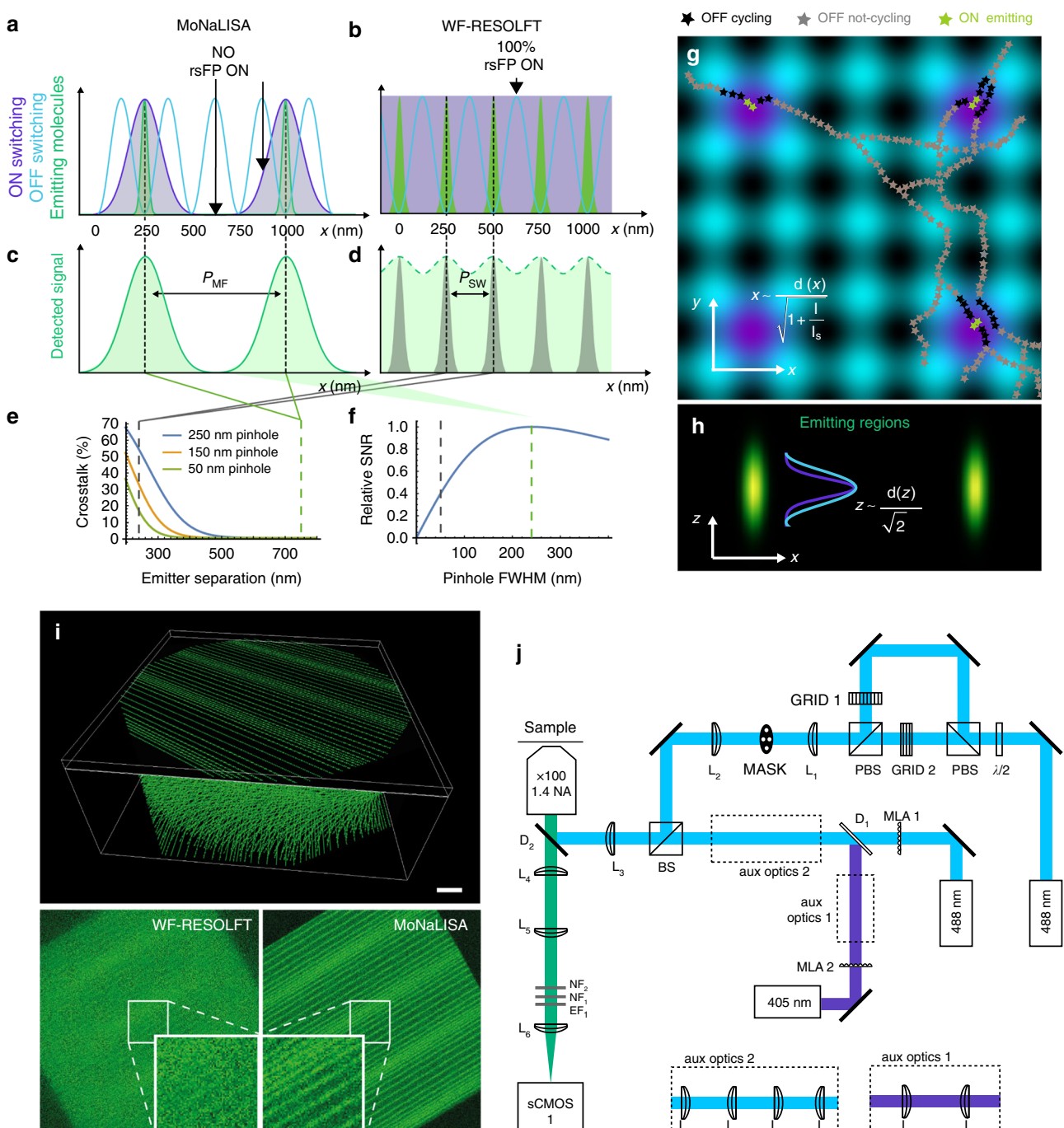

**Fig. 1** Basic concept and imaging scheme. 1D representation of MoNaLISA (**a**) and wide-field RESOLFT (**b**) illuminations. The ON-switching (violet) and the excitation are multiple Gaussian foci (periodicity $P_{MF}$) in MoNaLISA and homogenous in WF-RESOLFT. The OFF-switching light (blue, periodicity $P_{SW}$) confines the fluorescence emission (green). Detected fluorescence distributions for MoNaLISA (**c**) and WF-RESOLFT (**d**) as green area. **e** The larger foci separation $P_{MF} > P_{SW}$ in MoNaLISA allows detection of the whole fluorescence signal without crosstalk. **f** Fluorescence collection efficiency for different sizes of digital pinholing. MoNaLISA 250 nm optimal size (green dashed line) increases photon collection and SNR compared to the 50 nm pinholing of WF-RESOLFT (gray dashed line). **g** 2D calculation of the MoNaLISA illumination: ON-switching and excitation (violet), OFF-switching (blue). Filaments showing rsFPs in different states are illustrated. MoNaLISA minimizes ON–OFF cycling since most of rsFP facing OFF light maxima are in the OFF state. Scale bars, 250 nm. **h** The fluorescence confinement is extended in 3D as shown in the x–z section. Scale bars, 250 nm. **i** Simulations show that the combination of increased photon collection, saving rsFP switching cycles and 3D confinement of MoNaLISA images allow imaging of structures which are not observed in WF-RESOLFT. The simulated structure is composed of straight lines with varying separation (~80–300 nm) in planes separated by 300 nm along the optical axis. Scale bars, 1 μm (top), 2.5 μm (large bottom), and 500 nm (zoom inset). **j** Schematic representation of the optical set-up. MLA 1, MLA 2: microlens arrays, GRID 1, GRID 2: diffraction gratings, PBS polarizing beam splitter, BS non-polarizing beam splitter with 90/10 or 50/50 reflection/transmission, MASK custom-built mask to let through only the orders 1 and −1 of the two orthogonally polarized beams, NF notch filter, EF emission filter, and D dichroic mirror. L1: $f = 100$ mm, L2: $f = 300$ mm, L3: $f = 300$ mm, L4: $f = 250$ mm, L5: $f = 250$ mm, L6: $f = 200$ mm, L7: $f = 200$ mm. Aux optics 1. L9: $f = 200$ mm, L10: $f = 150$ mm, L11: $f = 250$ mm, L12: $f = 250$ mm. Aux optics 2. L7: $f = 100$ mm, L8: $f = 100$ mm

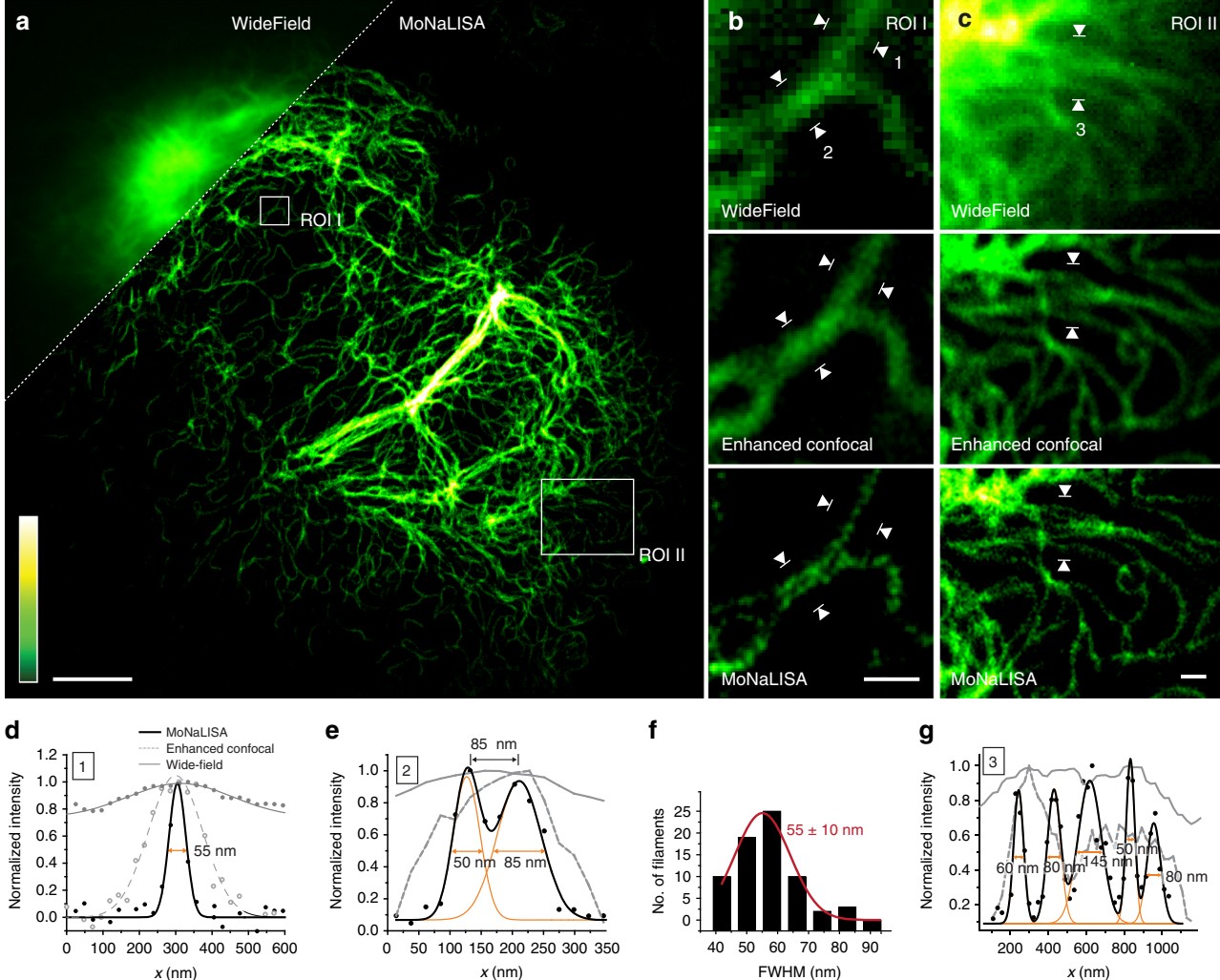

**Fig. 2** MoNaLISA nanoscale imaging. **a** Super-resolved image of endogenous vimentin-rsEGFP2 in comparison to the related wide-field image (inset). Scale bar 5 μm. **b** and **c** Magnified regions showing wide-field, enhanced confocal, and MoNaLISA images. Scale bar 500 nm. **d** Representative normalized intensity profiles (four lines averaged) measured across the filaments marked with white arrowheads no. 1 in **b**. Gaussian fits of the data show a FWHM of 55 ± 5 nm, 180 ± 10 nm, and 335 ± 45 nm for imaging in MoNaLISA, enhanced confocal, and wide-field mode, respectively (±s.e.m.). **e** Normalized intensity profiles (four lines averaged) measured across the filaments marked with white arrowheads no. 2 in **b**. The Gaussian fit for the two filaments shows a FWHM of 50 ± 5 nm and 85 ± 10 nm, respectively, and a separation of 85 ± 5 nm (±s.e.m.). **f** Histogram of 70 independent filament widths measured in image **a**, the mean value is 55 ± 10 nm (mean ± s.d.). **g** Normalized intensity profile measured in **f** across the white arrows. The MoNaLISA data shows five distinct filaments that cannot be resolved using confocal or widefield imaging

We first imaged a human knock-in cell line expressing endogenous level of vimentin-rsEGFP2 (see Fig. 2a)[23]. The image has been reconstructed from 900 raw images recorded by scanning an area of $750 \times 750$ nm$^2$ (foci to foci) with 25 nm steps. Each raw image is based on a switching cycle duration within 1.6–3.5 ms (see Supplementary Table 1). The reconstruction is simply used to assign the detected photons to a certain pixel and it does not force a priori conditions (see Supplementary Note 2). The improved spatial resolution is thus the result of the spatially confined emission of fluorescence and it is not imposed by processing. MoNaLISA imaging clearly resolved endogenous vimentin stretches with sizes down to 45–65 nm (see Fig. 2b–g), which would have been indistinguishable in WF and even in the enhanced confocal images acquired without OFF-switching. Importantly, the improved contrast enabled us to image small and dim filaments, which would have been missed with previous WF-RESOLFT implementations due to the lower SNR caused by inefficient photon collection (see Supplementary Figure 1C–D) and out-of-focus background.

**3D and 4D imaging in living cells and tissues**. The enhanced photon collection and minimal fatigue enabled to collect multiple optical sections of cells and tissues. Here, we demonstrate the optical sectioning performance of MoNaLISA by recording a stack of 28 super-resolved frames spaced 100 nm along the third dimension (see Fig. 3a). The optical sectioning of MoNaLISA is about 1.4 higher than in confocal microscopy (see Supplementary Note 1 and Supplementary Figure 4) and it can be easily extended to different cell lines and tissues. As an example of general applicability of the concept, we present multiple super-resolved z-stacks recorded in human epithelial cells (see Supplementary Movies 1–2), mouse astrocytes (see Supplementary Figure 10), organotypic tissues (see Fig. 3d), and embryonic stem cell (see Fig. 3e and Supplementary Movie 3). The super-resolution imaging ability is applied up to 15 μm from the growth surface and allowed to resolve actin bundles at resolutions of 50–70 nm, which stretched along the depth of the entire cell (see Supplementary Figure 10). The volumetric imaging was also extended over time to study the whole re-arrangement of the stem-cell

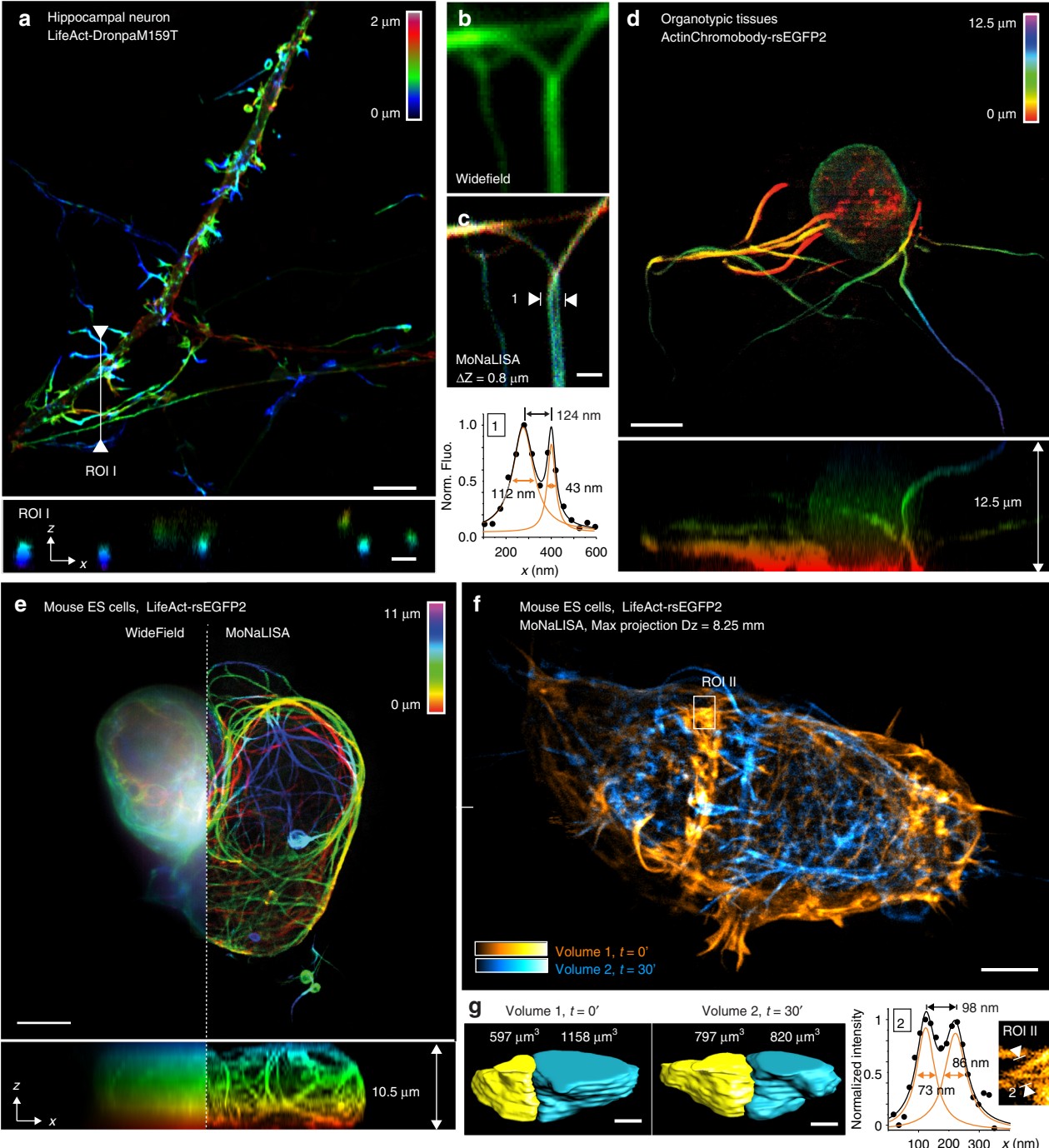

**Fig. 3** Optical sectioning and 3D recordings in living cells and organotypic tissues. **a** 3D stacks of 15 DIV neurons expressing LifeAct-DronpaM159T with MoNaLISA. The fine optical sectioning reveals the color-coded depth information in the *x–y* and *x–z* maximum projections. Scale bar, 5 μm. **b** and **c** Comparison between wide-field and MoNaLISA imaging of a neuron expressing LifeAct-rsEGFP2. The line profiles measured in the MoNaLISA image were averaged over four lines and fitted with a Gaussian. **d** Young brain cell recorded in an organotypic hippocampal rodent brain tissue and represented in an *x–y* and *x–z* maximum projection. Scale bars, 5 μm. **e** Colony of mouse ES cells spanning 30 × 30 × 11 μm³ imaged in the volumetric imaging mode. The scan steps are 35 nm in *x–y* and 500 nm in *z*. The images report fine structures across the entire *z* depth (right), hidden in the wide-field comparison (left). **f** Two volumes of mouse ES stem cells composed of 33 frames for a total axial depth of 8.25 μm has been recorded. The frame recording time was 2.9 s. We waited 30 min between volumes to observe the rearrangements of the actin network upon transfer to serum-free conditions. Scale bar, 5 μm. **g** The volumetric rendering of the two cells over time shows a clear shift in volume ratio in physiological (volume 1) versus the starving (volume 2) condition. ROI I is the magnified region of a single section and highlights fine actin bundles of sub-100 nm size

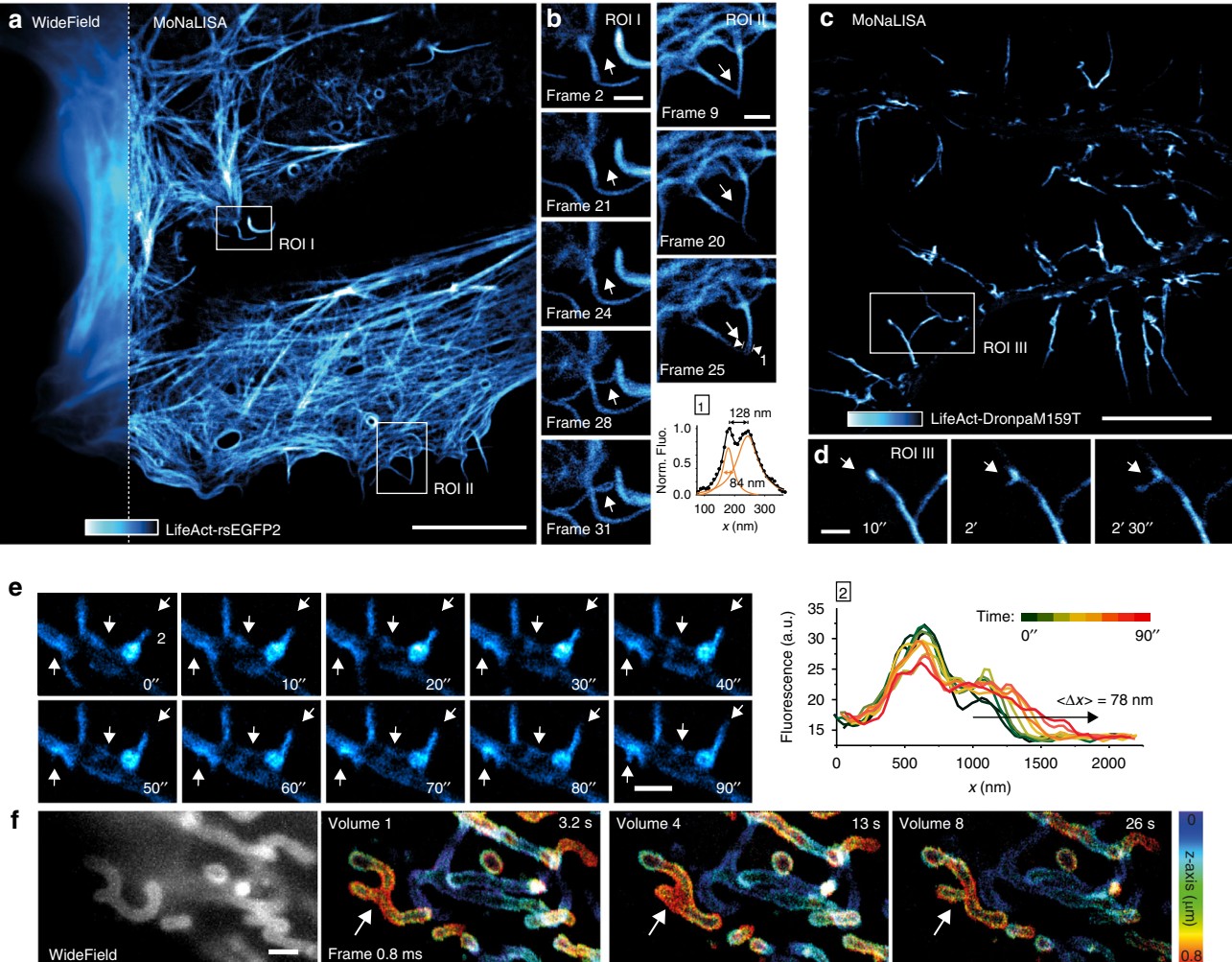

**Fig. 4** MoNaLISA time-lapse imaging. **a** Nanoscale imaging of epithelial cells expressing LifeAct-rsEGFP2 compared to the wide-field image (inset). Scale bar, 10 μm. **b** Magnified regions I and II unravel the dynamics of growing small filopodia observed over 3 min for a total of 45 frames, each acquired in 3.9 s. Scale bar, 1 μm. **c** Living hippocampal neurons infected with LifeAct-DronpaM159T showing the actin arrangements in dendrites and dendritic spines. Scale bar, 10 μm. **d** Magnified region III highlights the recruitment of actin molecules into a spine, which slowly enlarge its head into a cup-shape across 19 frames (left arrow), each acquired in 3.4 s. Scale bar, 1 μm. **e** Dynamics of dendritic filopodia emerging from actin densities in the dendritic shaft in neuron at 8 DIV. The arrows highlight the actin dynamics and redistribution. The graph shows the elongation over time of the filopodia 2 and reveals nanosized increments across time. Each line profile was averaged over a 100 nm width. **f** Nanoscale imaging of the mitochondria outer membrane labeled with rsEGFP2-Omp25. The mitochondrial volumetric dynamics are recorded continuously over seconds. The axial information is color-coded. Single frame recording 0.83 s, volume 3.3 s. Scale bar, 1 μm

colony (see Fig. 3f, g). Increased imaging depth can be achieved with the use of glycerol/water objective lenses with a correction collar to correct for spherical aberrations or dedicated adaptive optics[24,25]. The fluorescence signal gained in MoNaLISA, the reduced doses of light and the possibility to use genetically encoded markers for live cell imaging make MoNaLISA a suitable tool to study the rearrangement of the cells over time with nanoscale resolution and without compromising the viability of the cell (see Supplementary Figures 11 and 12). We provide multiple examples (see Fig. 4a–d and Supplementary Movies 4–7) of cellular time lapse over 30–50 super-resolved frames at low light levels. Structural changes in living epithelial cells (see Fig. 4a, b) and hippocampal neurons (see Fig. 4c–e) were unraveled by monitoring the actin dynamics in the time windows of seconds and minutes. Due to the enlarged field of view we could spot the formation of small actin loops (see Fig. 4b), the creation of a cup-shaped dendritic spines (see Fig. 4d), as well as the formation of actin branches at sub-80 nm step size (see Fig. 4e), which have been followed over time without loss in either spatial resolution

or image quality. Additionally, we applied MoNaLISA imaging to examine the continuous sliding of mitochondria above and below each other as super-resolved hollow organelles (see Fig. 4f and Supplementary Movie 8). Their fine changes in morphology during fusion and fission are caught in several sequential volumetric recordings across seconds.

## Discussion
The field of fluorescence microscopy has evolved rapidly and different techniques have achieved remarkable results in terms of spatial super-resolution, speed, light-dose reduction, contrast and optical sectioning in live-cell imaging.

Recent structured illumination implementations such as iSIM[26], are considerably faster than MoNaLISA reaching acquisition times of down to ~10 ms per frame. However, despite their impressive time resolution, they are essentially limited by diffraction and the gain in terms of spatial resolution is a factor of ~ $\sqrt{2}$ with respect to their diffraction-limited counterpart given the

same NA objective. For example, iSIM characterizations of spatial resolution report 213 ± 26 nm for raw images and 145 ± 14 nm after deconvolution for the lateral dimensions[26], which is 2–4 times worse than MoNaLISA imaging. Another implementation of the SIM concept in total internal reflection (TIRF-SIM)[27], when combined to ultra-high NA[18] and short wavelengths, report images acquired in seconds and with theoretically calculated spatial resolution approaching the sub-100 nm. Nevertheless, in these cases, like in any other TIRF technique, imaging is bound to the cover glass, limiting its application to membrane-related biological questions. In contrast, MoNaLISA can reach the entire cell volume, even in tissues. PA-NL-SIM[18] claims a theoretical resolution of 45 nm at a temporal resolution of 0.5–1 s, which are values very similar to the ones reported in our work. However, there are two major aspects that need to be taken into account: firstly, PA-NL-SIM is still based on a TIRF scheme meaning that, just like TIRF-SIM, it is intrinsically limited to imaging at the cover glass surface. Secondly, little or no experimental evidence[20] has been given of the theoretical 45 nm resolution. PA-NL-SIM in combination with the lattice light sheet illumination scheme named LLS-PA-NL-SIM[18] has been claimed to achieve a spatially anisotropic super-resolution of $118 \times 230 \times 170$ nm (x, y, z). Thus, this implementation shows true non-linear super-resolution in only one dimension. The LLS-PA-NL-SIM acquisition time for a 1D super-resolution improvement is about 2.5 s (a volume containing 28 slices is reportedly imaged in 70.5 s), which is comparable and even longer than in MoNaLISA imaging.

STED[28] is potentially the fastest diffraction-unlimited super resolution methods and it has been demonstrated to work in living samples even at video rate[29]. However, the recording speed can only be achieved in a small field of view because of the point-scanning implementation. A recording time of a field of view of $45 \times 45$ $\mu m^2$ in STED would take about 20–40 times longer than in MoNaLISA parallelized recording. A parallelized version of STED has been reported[30,31] but, to our knowledge, it has not been demonstrated in living samples and time-lapse recordings. Additionally, since STED is based on the stimulated emission depletion mechanism, recordings require light intensities (MW-GW cm$^{-2}$) which are notoriously several orders of magnitude higher than the ones used in MoNaLISA (W-kW cm$^{-2}$).

Single molecule switching methods, such as PALM/STORM/FPALM[2–4] and the 3D-derived technique such as iPALM[32–34], 3D STORM based on PSF astigmatism[35], multi-planes[36] or other PSF engineering methods[37] are very effective in terms of spatial resolution but they compromise recording speed and that is why most of these experiments are carried out with fixed samples. There have been efforts to apply single molecule switching methods to live-cell imaging, for example 3D STORM was successfully applied to image clathrin-coated pits at 1–2 s temporal resolution[38,39]. However, structures which need higher labeling density to be resolved will inevitably compromise the recording speed, mostly because of the intrinsic trade-off in the concept between spatial resolution, which requires a high number of localizations, and acquisition time.

Finally, WF-RESOLFT[13] can achieve temporal resolutions of ~1 s per frame which is comparable to MoNaLISA, however it lacks optical sectioning and it is much less efficient in photon collection, which decreases the image contrast and overall quality. In fact, an increase in the length of the switching cycle has been reported in order to improve the SNR at the cost of reducing the temporal resolution to ~17 s per frame[14]. Moreover, neither WF-RESOLFT nor PA-NL-SIM, which are both based on rsFP, have been demonstrated to our knowledge at endogenous levels of fluorescent proteins expression.

To sum up, while different existing techniques are as good as, or better than MoNaLISA in one or some metrics regarding live-

cell, super-resolution imaging, no technique to date presents at the same time the advantages of our approach in terms of minimal light doses, optical sectioning, field of view size, spatial resolution, and flexibility to visualize structures in the entire cell.

We have demonstrated that MoNaLISA enables entire living cells and tissues to be studied in 4D with a combination of high spatio-temporal resolution and minimal doses of light. The efficient use of ON–OFF switching cycles in combination with intrinsic optical sectioning and optimal photon collection enabled the imaging of subcellular nanoscale structures at endogenous, physiologically relevant expression levels, which would have been missed in previous implementations. The collected data are in good agreement with simulations and theory, which confirmed the improvements in image quality. The time resolution did not yet reach its theoretical limit. Faster camera technology, as well as condensed acquisitions in a single camera exposure as in instant SIM[26] have the potential to maximize the recording speed.

Finally, MoNaLISA advances several aspects in the fields, such as image contrast, speed, and high throughput with larger fields of view. Hence, it paves the way for molecular and cell biology studies of living cells and tissues in 4D that were previously inaccessible.

## Methods

**MoNaLISA set-up**. MoNaLISA set-up is described in Fig. 1h. The 405 nm and the 488 nm light are generated by laser-diodes digitally modulated (200 mW, 06-01, 405 nm and two separate 200 mW, 06-01, 488 nm, Cobolt). The output of the three lasers are first spectrally cleaned up with a matched bandpass filter (ZET 405/10 and ZET 488/10, Chroma) and then expanded with a telescope (AC254-30-A-ML and AC254-300-A-ML, Thorlabs). The beam profiles are spatially cleaned up with a pinhole (P20S, Thorlabs) in the telescope intermediate focus. Both the 405 nm ON-switching and 488 nm fluorescence excitation collimated beams are focused by a microlens array (MLA-150-5C-M, Thorlabs) that creates the multi-foci pattern. Subsequently, for the 405 nm light path an auxiliary telescope (Aux optics 1) is used to correct for a chromatic mismatch in magnification. The corrected 405 nm and the 488 nm multi-foci patterns are then combined in the same beam path by a dichroic mirror (ZT458RDC, Chroma). A second set of telescopes (Aux optics 2) is used as relay optics but also to adjust the desired foci to foci periodicity and beamlet size. We used Thorlabs AC254-300-A-ML and AC254-400-A-ML to obtain a period of $P_{MF} = 750$ nm. This intermediate plane is then imaged into the sample by the focusing lens (AC508-300-A-ML, Thorlabs) and the objective (HCX PC APO 100×/1.40-0.70 oil, Leica Microsystems) creating a multi-foci pattern illumination.

The OFF-switching pattern is adapted from the WF-RESOLFT implementation[13] since it provides a robust array of intensity zeros while keeping simplicity in the instrumentation. The collimated light is directed through a half-wave plate (B. Halle, Germany) and a polarizing beam splitter (Thorlabs CCM1-PBS251/M). The half-wave plate is adjusted to equally split the light according to the p and s orientation. After the PBS, the light is sent through custom-made phase-diffraction gratings of 437 nm-high SiO$_2$ lines with a 25 $\mu m$ period (Laser Laboratorium Göttingen). The gratings were oriented such that the grid lines were parallel to the polarization of the incoming light in order to ensure s-polarization at the focal plane of objective. After the gratings, the light paths were recombined with another polarizing beam splitter (Thorlabs CCM1-PBS251/M). The gratings were placed in the back focal plane of an $f = 100$ mm lens (AC254-80-A-ML, Thorlabs), the diffracted light was then focused to a mask where all of the diffraction orders except +1 and −1 were blocked; these orders were relayed by a pair of lenses (AC508-300-A-ML, Thorlabs) to four spots in the back focal plane of the objective, where they recombined and created a regular sinusoidal pattern in two orientations with a period $P_{SW} = 250$ nm. The period of the sinusoidal pattern can be tuned by changing the focal length of the first lens after the grids. To combine the ON-switching/excitation light path with the OFF-switching light path we use a 90/10 non-polarizing beam splitter (CCM1-BS028/M, Thorlabs).

The ON-switching, excitation and OFF-switching beam paths are reflected with a dichroic (ZT488RDC, Chroma) into the objective. The fluorescence light coming from the sample is transmitted by the same dichroic and an auxiliary telescope (L4, L5) is used as relay optics. In one-color operation, light is then directed through 488 and 405 notch filters (ZET405NF and ZET488NF, Chroma), a bandpass fluorescence emission filter (ET535/70, Chroma) and finally imaged into an sCMOS camera (ORCA Flash 4.0 V2, Hamamatsu) by an $f = 200$ mm lens (AC254-200-A-ML, Thorlabs).

For sample screening we added a wide field path, which can be alternated with the main MoNaLISA imaging illumination by flipping a motorized mirror. For ON-switching and excitation we use 405 nm (BDL-405 Picosecond Diode Laser, Becker and Hickl) and 473 nm laser light (50 mW, 04-01, 473 nm, Cobolt), which

are collimated and expanded by a telescope (AC508-30-A-ML and AC508-250-A-MLThorlabs) and then focused into the back focal plane of the objective by the same lens used in the previously described multi-foci optical path (AC508-300-A-ML, Thorlabs).

**Image acquisition**. MoNaLISA imaging is based on a three steps illumination and piezo scanning scheme (Nanomax-TS, Thorlabs). The ON-switching, excitation and OFF-switching lasers light are digitally modulated by TTL signals with a NIDAQ PCI 6371 (National Instrument) acquisition device. The NIDAQ synchronizes the pulse sequence, the camera read-out and the scanning. The readout laser is synchronized with the sCMOS camera exposure to detect fluorescent emission. The length and delay of the laser pulses, the output power of the lasers, the exposure time of the camera and all other relevant parameters can be controlled through our graphical user interface which in turn is part of our own open-source microscope control project developed in Python. Our free software, initially based on *Tormenta*[40], is an effort to contribute to the open-source community and it provides an alternative to commercial software available for the control of a scanning microscope. It is constantly under development with the current configuration available at https://github.com/TestaLab. The software uses adaptions of available instrument drivers generously contributed by other members of the open-source community https://github.com/pearu/pylibnidaqmx, https://github.com/ZhuangLab/storm-control/tree/master/sc_hardware/hamamatsu, https://github.com/LabPy/lantz.

After image acquisition the data file is processed in our custom image reconstruction software. The software quantifies the amount of emission from each emitter in each step of the imaging scan through a digital pinholing process. The quantified signal is then re-assigned to form the final image. We applied bleaching correction between frames to compensate for switching fatigue and we correct for further salt and pepper noise found in our sCMOS camera. A detailed description of the full reconstruction procedure is provided in the supplementary information (see Supplementary Note 2–3).

All the reported images are raw data, to help visualization we either applied a Gaussian smoothing of 20 nm (see Figs. 2, 4 and 3e) or an up-sampling with a bilinear interpolation (see Fig. 4).

**Epithelial cell culture and rsFPs plasmids**. U2OS (ATCC® HTB-96™) and PtK2 (Sigma-Aldrich, 88031601) cells were cultured in Dulbecco's modified Eagle medium (DMEM) (Thermo Fisher Scientific, 41966029) and Eagle's minimum essential medium (MEM) (Sigma Aldrich, M2279) supplemented with 10% (vol/vol) fetal bovine serum (Thermo Fisher Scientific, 10270106), 1% Penicillin–Streptomycin (Sigma Aldrich, P4333) and maintained at 37 °C and 5% $CO_2$ in a humidified incubator. For transfection, $2 \times 10^5$ cells per well were seeded on coverslips in a six-well plate. After one day cells were transfected using Lipofectamine LTX Reagent with PLUS reagent (Thermo Fisher Scientific, 15338100) according to the manufacturer's instructions. 24–36 h after transfection cells were washed in phosphate-buffered saline (PBS) solution, placed with phenol-red free DMEM or Leibovitz's L-15 Medium (Thermo Fisher Scientific, 21083027) in a chamber and imaged at 35 °C. The plasmid rsEGFP2-Omp25 was cloned by replacing the HaloTag coding region from Halo-Omp25[41] with the coding sequence for rsEGFP2 via XbaI/SpeI restriction digest. The plasmid Halo-Omp25 was kindly gifted by the Dr. Rothman laboratory. The LifeAct-rsEGFP2, LifeAct-DronpaM159T, LifeAct-rsEGFPN205S, and the endogenously tagged vimentin-rsEGFP2 cell lines[23] were a kind gift from Dr. Stefan W. Hell and Dr. Stefan Jakobs (MPI-BCP Göttingen, Germany).

**Hippocampal neuron cultures**. Primary hippocampal cultures were prepared from embryonic day E18 Sprague Dawley rat embryos. Hippocampi were dissected and mechanically dissociated in MEM. $20 \times 10^3$ cells per well were seeded in 12-well plates on a poly-D-ornithine (Sigma Aldrich, P8638) coated #1.5 18 mm (Marienfeld) glass coverslips and let them attach in MEM with 10% horse serum (Thermo Fisher Scientific, 26050088), 2 mM L-Glut (Thermo Fisher Scientific, 25030-024), and 1 mM sodium pyruvate (Thermo Fisher Scientific, 11360-070). After 3 h the media was changed with Neurobasal (Thermo Fisher Scientific, 21103-049) supplemented with 2% B-27 (Thermo Fisher Scientific, 17504-044), 2 mM L-glutamine and 1% penicillin–streptomycin. The experiments were performed on cultures from DIV3 to DIV22. Young neurons (<10DIV) were transfected with the plasmid LifeAct-rsEGFP2 using Lipofectamine 2000 transfection reagent (Thermo Fischer Scientific, 11668027) following the manufacturer's instructions. For older cultures, 12–24 h before experiments, the cells were infected with a modified Semliki forest virus expressing either the actin-binding protein LifeAct together with the photo-switchable protein DrompaM159T or the actin-chromobody together with the photo-switchable protein rsEGFP2, by adding 1 µL of the virus to the culture medium. All experiments were performed in accordance with animal welfare guidelines set forth by Karolinska Institutet and were approved by Stockholm North Ethical Evaluation Board for Animal Research.

**Embryonic stem cell preparation**. Mouse R1 embryonic stem cells (ESCs) were obtained from Sigma-Aldrich (07072001-1VL) and cultivated on a layer of gamma-irradiated CF-1 mouse embryonic fibroblasts (Tebu-Bio, GSC-6201G) in DMEM formulated with high glucose and sodium pyruvate and supplemented with 15% FBS, $10^3$ units/mL LIF, 1% β-mercaptoethanol, 100 units/mL penicillin, 100 µg/mL streptomycin and maintained at 37 °C and 5% $CO_2$ in a humidified incubator. For transfection, $2 \times 10^5$ ESCs were incubated with 800 ng of plasmid for 15 min in suspension[42]. Imaging was done within 24–48 h following transfection. In the starvation experiment, the medium was replaced by OptiMEM reduced serum (Thermo Fisher Scientific, 31985-062).

**U2OS and astrocyte cell culture**. U2OS cells and C8-D1A astrocytes were cultivated in Dulbecco's modified Eagle medium (DMEM) and Ptk2 cells were cultivated in Eagle's minimum essential medium (MEM). Both DMEM and MEM were supplemented with 10% (vol/vol) FBS, 100 units/mL penicillin, 100 µg/mL streptomycin and cells maintained at 37 °C and 5% (vol/vol) $CO_2$ in a humidified incubator.

For transfection, cells were seeded on coverslips in a six-well plate. After one day cells were transfected with Lipofectamine LTX Reagent with PLUS reagent (Life Technologies, Carlsbad, USA) according to the manufacturer's instructions. 24–48 h after transfection cells were washed and mounted with phenol red-free DMEM (Invitrogen, Carlsbad, USA) onto concavity slides. To prevent samples from drying, the coverslips were sealed with twinsil (Picodent, Wipperfürth, Germany). Finally, the cells were imaged at room temperature. Cells were mounted in phenol red-free DMEM (Invitrogen) and imaged 24–72 h post transfection.

**Organotypic brain slice preparation and infection**. Hippocampal brain slices were prepared by dissecting hippocampi from postnatal day P6–P16 mice, which were then sectioned with a Tissue Chopper (MC ILWAIN) in 400 µm thick slices and plated on a poly-D-ornithine-coated #1.5 18 mm glass coverslips. The slices were maintained in a humidified incubator at 37 °C in MEM supplemented with 25% HBSS (Thermo Fisher Scientific, 14175053), 25% horse serum, 1% L-glutamine, 1% sodium pyruvate, 1% penicillin–streptomycin.

Slice cultures were infected three days after plating. For infection, a modified Semliki forest virus expressing either the actin-binding protein LifeAct together with the photo-switchable protein DronpaM159T or the actin-chromobody together with the photo-switchable protein rsEGFP2 was used.

1 µL of the virus was added in the media where the slices were cultured. The cultures were then incubated for at least 12 h and imaged within 12–48 h after infection. For imaging, the coverslips with the brain slices were transferred to an imaging chamber and maintained in artificial cerebrospinal fluid (ACSF). The imaging chamber and the objective lens were generally heated to 35 °C during the experiments.

**Code availability**. The software used to acquire the super-resolved data are freely available for academic use and they are provided as open source Python code in github as https://github.com/TestaLab

**Data availability**. The data that support the findings of this study are available from the corresponding author upon reasonable request.

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

# ARTICLE

10. Habuchi, S. et al. Reversible single-molecule photoswitching in the GFP-like fluorescent protein Dronpa. *Proc. Natl Acad. Sci. USA* **102**, 9511–9516 (2005).

11. Grotjohann, T. et al. rsEGFP2 enables fast RESOLFT nanoscopy of living cells. *eLife* **1**, e00248 (2012).

12. Brakemann, T. et al. A reversibly photoswitchable GFP-like protein with fluorescence excitation decoupled from switching. *Nat. Biotechnol.* **29**, 942–947 (2011).

13. Chmyrov, A. et al. Nanoscopy with more than 100,000 'doughnuts'. *Nat. Methods* **10**, 737–740 (2013).

14. Chmyrov, A. et al. Achromatic light patterning and improved image reconstruction for parallelized RESOLFT nanoscopy. *Sci. Rep.* **7**, 44619 (2017).

15. Gustafsson, M. G. L. Nonlinear structured-illumination microscopy: Wide-field fluorescence imaging with theoretically unlimited resolution. *Proc. Natl Acad. Sci. USA* **102**, 13081–13086 (2005).

16. Heintzmann, R., Jovin, T. M. & Cremer, C. Saturated patterned excitation microscopy - a concept for optical resolution improvement. *J. Opt. Soc. Am. A* **19**, 1599–1609 (2002).

17. Rego, E. H. et al. Nonlinear structured-illumination microscopy with a photoswitchable protein reveals cellular structures at 50-nm resolution. *Proc. Natl Acad. Sci. USA* **109**, E135–E143 (2012).

18. Li, D. et al. Extended-resolution structured illumination imaging of endocytic and cytoskeletal dynamics. *Science* **349**, aab3500 (2015).

19. Zhang, X. et al. Highly photostable, reversibly photoswitchable fluorescent protein with high contrast ratio for live-cell superresolution microscopy. *Proc. Natl Acad. Sci. USA* **113**, 10364–10369 (2016).

20. Sahl, S. J. et al. Comment on "Extended-resolution structured illumination imaging of endocytic and cytoskeletal dynamics". *Science* **352**, 527 (2016).

21. Bewersdorf, J., Pick, R. & Hell, S. W. Multifocal multiphoton microscopy. *Opt. Lett.* **23**, 655–657 (1998).

22. York, A. G. et al. Resolution doubling in live, multicellular organisms via multifocal structured illumination microscopy. *Nat. Methods* **9**, 749–754 (2012).

23. Ratz, M., Testa, I., Hell, S. W. & Jakobs, S. CRISPR/Cas9-mediated endogenous protein tagging for RESOLFT super-resolution microscopy of living human cells. *Sci. Rep.* **5**, 9592 (2015).

24. Booth, M. J., Neil, M. A. A., Juskaitis, R. & Wilson, T. Adaptive aberration correction in a confocal microscope. *Proc. Natl Acad. Sci. USA* **99**, 5788–5792 (2002).

25. Testa, I. et al. Nanoscopy of living brain slices with low light levels. *Neuron* **75**, 992–1000 (2012).

26. York, A. G. et al. Instant super-resolution imaging in live cells and embryos via analog image processing. *Nat. Methods* **10**, 1122–1126 (2013).

27. Fiolka, R., Beck, M. & Stemmer, A. Structured illumination in total internal reflection fluorescence microscopy using a spatial light modulator. *Opt. Lett.* **33**, 1629–1631 (2008).

28. Hell, S. W. & Wichmann, J. Breaking the diffraction resolution limit by stimulated emission: stimulated-emission-depletion fluorescence microscopy. *Opt. Lett.* **19**, 780–782 (1994).

29. Westphal, V. et al. Video-rate far-field optical nanoscopy dissects synaptic vesicle movement. *Science* **320**, 246–249 (2008).

30. Yang, B., Przybilla, F., Mestre, M., Trebbia, J. B. & Lounis, B. Large parallelization of STED nanoscopy using optical lattices. *Opt. Express* **22**, 5581–5589 (2014).

31. Bergermann, F., Alber, L., Sahl, S. J., Engelhardt, J. & Hell, S. W. 2000-fold parallelized dual-color STED fluorescence nanoscopy. *Opt. Express* **23**, 211–223 (2015).

32. Shtengel, G. et al. Interferometric fluorescent super-resolution microscopy resolves 3D cellular ultrastructure. *Proc. Natl Acad. Sci. USA* **106**, 3125–3130 (2009).

33. Aquino, D. et al. Two-color nanoscopy of three-dimensional volumes by 4Pi detection of stochastically switched fluorophores. *Nat. Methods* **8**, 353–359 (2011).

34. Huang, F. et al. Ultra-high resolution 3D imaging of whole cells. *Cell* **166**, 1028–1040 (2016).

35. Huang, B., Wang, W., Bates, M. & Zhuang, X. Three-dimensional super-resolution imaging by stochastic optical reconstruction microscopy. *Science* **319**, 810–813 (2008).

36. Juette, M. F. et al. Three-dimensional sub-100 nm resolution fluorescence microscopy of thick samples. *Nat. Methods* **5**, 527–529 (2008).

37. Pavani, S. R. et al. Three-dimensional, single-molecule fluorescence imaging beyond the diffraction limit by using a double-helix point spread function. *Proc. Natl Acad. Sci. USA* **106**, 2995–2999 (2009).

38. Huang, F. et al. Video-rate nanoscopy using sCMOS camera-specific single-molecule localization algorithms. *Nat. Methods* **10**, 653–658 (2013).

39. Jones, S. A., Shim, S. H., He, J. & Zhuang, X. Fast, three-dimensional super-resolution imaging of live cells. *Nat. Methods* **8**, 499–508 (2011).

40. Barabas, F. M., Masullo, L. A. & Stefani, F. D. Note: Tormenta: an open source Python-powered control software for camera based optical microscopy. *Rev. Sci. Instrum.* **87**, 126103 (2016).

41. Takakura, H. et al. Long time-lapse nanoscopy with spontaneously blinking membrane probes. *Nat. Biotechnol.* **35**, 773–780 (2017).

42. Tamm, C., Kadekar, S., Pijuan-Galito, S. & Anneren, C. Fast and efficient transfection of mouse embryonic stem cells using non-viral reagents. *Stem Cell Rev. Rep.* **12**, 584–591 (2016).

## Acknowledgements

We thank the European ERC starting grant 'MoNaLISA' funding http://dx.doi.org/10.13039/501100000781 for supporting the project. G.C. and I.T. thank the Swedish Research Council for funding. J. Alvelid and F. Barabas are acknowledged for critical reading. L. Gustafsson is acknowledged for helping with the image quantification.

## Author contributions

I.T. conceived idea, designed the optical system and supervised research. L.A.M., A.B., F.P. built the optical system and implemented data acquisition software. I.T., L.A.M., A.B., F.P. acquired data. F.P., G.C. and M.R. prepared samples. All authors analyzed data. I.T. wrote the manuscript with input from all the authors.

## Additional information

**Competing interests:** The authors declare no competing interests.

