## [Peer Review File · Nature Communications]

Reviewers' comments:

Reviewer #1 (Remarks to the Author):

The authors have satisfactorily addressed all the concerns I raised in my previous revision for Nature Photonics. The paper is now more complete and the description of the proposed methodology is easier to follow.

Comment:

The overall position of the proposed method within the context of super-resolution would be highly welcome.

Reviewer #4 (Remarks to the Author):

Masullo et al. present MoNaLISA, a super-resolution microscope technique based on an advanced and technically adapted implementation of the RESOLFT concept. In my opinion, the manuscript is well written and presents the results in a comprehensive way. Many points raised by the referees were addressed in the revised manuscript. In particular, I believe the topic of phototoxicity has extensively been discussed. My only remaining problem is the topic of resolution estimates. I agree with the two previous referee comments that a resolution estimate that is based on sparse and subjective selections of filament structures is far from ideal. In my opinion, better resolution estimates are derived from imaging of well defined sub-resolution-sized structures like Origami or bead samples. This might be problematic here due to the use of reversibly switchable fluorescent proteins. However, the second well established criterion, Fourier ring correlation, should be well applicable also to this technique. I could not follow the reasoning of the authors against this method. Some dynamic processes are surely slow enough to get a resolution from two successive images. And possibly, also a fixed sample could be prepared in which the rsFP are still active. A FRC resolution estimate should accompany the given estimates from filament structures. Overall, I find the manuscript of great interest to a broad audience and recommend publication.

Response Letter to the reviewers comments

We thank the editor and reviewers for the valuable comments.

In the next section we provide a point-by-point response. The reviewers comments are in blue, our responses are in black and any additions to the main text and supplementary information is cited and explained below.

Reviewers' comments:

Reviewer #1 (Remarks to the Author):

The authors have satisfactorily addressed all the concerns I raised in my previous revision for Nature Photonics. The paper is now more complete and the description of the proposed methodology is easier to follow.

We thank the reviewer for supporting the publication of our work.

Comment:

The overall position of the proposed method within the context of super-resolution would be highly welcome.

Following the reviewer and editor's suggestion, we have now added to the manuscript the position of MoNALISA as a new method in the super resolution field. We added the information as "discussion" in a new paragraph of the main text. For completeness, we have also included some techniques not discussed previously, including MINFLUX and the very recently published TIRF-iSIM.

Reviewer #4 (Remarks to the Author):

Masullo et al. present MoNaLISA, a super-resolution microscope technique based on an advanced and technically adapted implementation of the RESOLFT concept. In my opinion, the manuscript is well written and presents the results in a comprehensive way. Many points raised by the referees were addressed in the revised manuscript. In particular, I believe the topic of phototoxicity has extensively been discussed.

We thank the reviewer for the positive comments.

My only remaining problem is the topic of resolution estimates. I agree with the two previous referee comments that a resolution estimate that is based on sparse and subjective selections of filament structures is far from ideal. In my opinion, better resolution estimates are derived from imaging of well defined sub-resolution-sized structures like Origami or bead samples. This might be problematic here due to the use of reversibly switchable fluorescent proteins. However, the second well established criterion, Fourier ring correlation, should be well applicable also to this technique. I could not follow the reasoning of the authors against this method. Some dynamic processes are surely slow enough to get a resolution from two successive images. And possibly, also a fixed sample could be prepared in which the rsFP are still active. A FRC resolution estimate should accompany the given estimates from filament structures. Overall, I find the manuscript of great interest to a broad audience and recommend publication.

We understand the reviewer concern on the resolution estimate based on subjective selections of filaments and line profiles. We minimized the degree of subjectiveness

by measuring the FWHM over 50 filaments in the same field of view/cell (Figure2 and S1). However, we choose on purpose the smallest filaments, since the final image is always the convolution of the point spread function with the structure in the specimen. Also each filaments is averaged over 4 consecutive lines to minimize noise.

We also agree with the reviewer that the Rayleigh principle based on the separation of two structures would provide additional information, especially regarding the signal to noise ratio. For that we have a few representative line profiles (Figure 2e and 2g) showing the ability to separate distinct filaments close to each other.

Even more informative would be to have objects of known size and positioned at a known distance as for DNA origami nanorulers.

Nanorulers conjugated with optimal dyes for STED, STORM and PAINT and SIM are easy to find and commercially available while as correctly pointed out by the reviewer, beads samples and origami with rsFPs are not commercially available and not easy to produce either. One reason is that the rsFPs switching properties are sensitive to fixation and embedding medium, resulting in a decrease in switching cycles and higher background.

[REDACTED]

[REDACTED]

[REDACTED]

As requested by the reviewer the possibility to use Fourier ring correlation (FRC) to quantify resolution of the system has been investigated.

So far, FRC analysis was successfully used and thoroughly characterized for PALM/STORM microscopy¹ and point-scanning systems² (such as STED or confocal microscopy). Used in this context, FRC analysis has been shown to be a robust tool for comparing the effective resolution of different imaging systems. As pointed out and demonstrated by Tortarolo et al. though, values obtained by FRC analysis generally underestimate the resolution if compared to values obtained using the Rayleigh criterion, especially if applied to raw data images, which are subjected to shot-noise and intrinsic detector noise.

Importantly, in PALM/STORM the “noise” of the reconstructed image is mainly given by the localization uncertainty whilst in point-scanning microscopy such as STED the noise of the image is a combination of the shot-noise and of the intrinsic noise of the single-point detector (APD, SPAD or PMT).

As for images acquired using camera-based systems (wide-field, SIM, iSIM) there has, to our knowledge, been little or no attempts to quantify resolution using FRC and there are several reasons why camera-based systems add an additional level of complexity to a potential FRC analysis. The main reason is that cameras exhibit several noise characteristics that point detectors don't, such as fixed pattern noise and pixel to pixel

variations. These properties of the sensor might introduce frequency components that are not easily separated from the frequencies of the image and that thus compromise the reliability of the results. On top of this, most camera based super resolution systems include some sort of image reconstruction. This process may also to some extent corrupt the frequency spectra of the final image in ways that are not trivial to take into account. Having said this, we agree with the many FRC advocates that a more robust and objective resolution evaluation standard is much needed. We thus assess MoNaLISA performance using FRC and found a consistent resolution improvement when comparing wide-field, enhanced confocal and MoNaLISA imaging (Figure B). In agreement with the results of Tortarolo et al. though, the numbers obtained are also consistently larger than what the analysis of line profile FWHM and Rayleigh criterion shows. Though the FRC analysis gives a resolution of 122 nm in the MoNaLISA images, we see in the same image that we can clearly resolve two filaments separated by $\sim 80-90$ nm. In the same way as has been shown for STED imaging, even if the FRC analysis can in principle be applied, the values obtained do not seem to correspond to what spatial separations that the system is able to resolve. For STED images to provide an FRC curve with a value at the $1/7$ cutoff below 100 nm the pixel photon counts are usually within several hundreds. This is possible because the specimens are labeled with multiple organic dyes, which are already several factors brighter than rsFPs. Also, the two images are usually recorded sequentially pixel by pixel (10-100 ns) in fixed not moving samples for STED, which minimizes potential loss of correlation due to motion. While the use of rsFPs strictly rely on live specimen and the images are recorded sequentially frame by frame with a larger delay of 0.5-1 second, which increase motion artifacts.

We believed, that future designed origami samples with higher load of rsFPs will allow to record two consecutive frames of an immobile system with higher SNR and smaller sampling and therefore to perform a more robust FRC estimation of MoNaLISA images.

We would happily engage in further discussions as to the use and characterization of FRC analysis in camera-based systems but believe this would be an extensive work on its own.

Figure B. (a) Actin labeled structures measured in Wide-Field, Enhance Confocal and MoNaLISA imaging modalities. The imaging pulse sequence includes 500 μs of ON-switching at 405 nm at 0.7 kW/cm², OFF-switching at 0.6 kW/cm² for 1.5 ms and 1.5 ms of read-out at 488 nm at 1.5 kW/cm². The geometry of the MoNaLISA microscope was 750 nm for the ON-switching and read-out light and 250 nm of OFF-switching periodicity, with a scan step of 25 nm. Scale bar, 10 μm (b) Enlargement of a region of interest showing the different spatial details in the widefield, the enhanced confocal and in the two consecutive MoNaLISA recordings. Scale bar, 1 μm . (c) FRC estimation of an image recorded with Wide-Filed (grey), Enhanced Confocal (green) and MoNaLISA (blue) modality.

We look forward to hearing from you regarding our submission. We would be glad to respond to any further questions and comments that you may have.

References:

1. Banterle N, Bui KH, Lemke EA, Beck M. Fourier ring correlation as a resolution criterion for super-resolution microscopy. *Journal of structural biology* **183**, 363-367 (2013).
2. Tortarolo G, Castello M, Diaspro A, Koho S, Vicidomini G. Evaluating image resolution in stimulated emission depletion microscopy. *Optica* **5**, 32-35 (2018).

REVIEWERS' COMMENTS:

Reviewer #4 (Remarks to the Author):

The authors have done a very good job in revising the manuscript and replying to all comments. I believe this is a nice manuscript ready for publication.